# Effects of Urban Vibrancy on an Urban Eco-Environment: Case Study on Wuhan City

**DOI:** 10.3390/ijerph19063200

**Published:** 2022-03-09

**Authors:** Ruijing Yu, Chen Zeng, Mingxin Chang, Chanchan Bao, Mingsong Tang, Feng Xiong

**Affiliations:** 1Department of Land Management, Huazhong Agricultural University, Wuhan 430070, China; yurj331@126.com (R.Y.); hzauggglcmx@126.com (M.C.); krybcc@163.com (C.B.); yevat@webmail.hzau.edu.cn (M.T.); 2Research Center for Territorial Spatial Governance and Green Development, Huazhong Agricultural University, Wuhan 430070, China; 3Sino-Ocean Group Holding Limited, Wuhan 430021, China; 13407123069@139.com

**Keywords:** urban vibrancy, urban eco-environment, spatial modelling, Wuhan

## Abstract

In the context of rapid urbanisation and an emerging need for a healthy urban environment, revitalising urban spaces and its effects on the urban eco-environment in Chinese cities have attracted widespread attention. This study assessed urban vibrancy from the dimensions of density, accessibility, liveability, diversity, and human activity, with various indicators using an adjusted spatial TOPSIS (technique for order preference by similarity to an ideal solution) method. The study also explored the effects of urban vibrancy on the urban eco-environment by interpreting PM 2.5 and land surface temperature using “big” and “dynamic” data, such as those from mobile and social network data. Thereafter, spatial modelling was performed to investigate the influence of urban vibrancy on air pollution and temperature with inverted and extracted remote sensing data. This process identified spatial heterogeneity and spatial autocorrelation. The majority of the dimensions, such as density, accessibility, liveability, and diversity, are negatively correlated with PM 2.5, thereby indicating that the advancement of urban vibrancy in these dimensions potentially improves air quality. Conversely, improved accessibility increases the surface temperature in most of the districts, and large-scale infrastructure construction generally contributes to the increase. Diversity and human activity appear to have a cooling effect. In the future, applying spatial heterogeneity is advised to assess urban vibrancy and its effect on the urban eco-environment, to provide valuable references for spatial urban planning, improve public health and human wellbeing, and ensure sustainable urban development.

## 1. Introduction

The urban eco-environment has attracted increasingly widespread concern in the context of rapid urbanisation and low-density urban expansion over the past decade. The eco-environment system is composed of all kinds of natural resources that human beings rely on for survival, including water, land and atmosphere [1]. Environmental problems such as air and water pollution, uncontrolled land use, heat island effect and resource scarcity have emerged, which are closely related to human health. Chronic exposure to air pollution, especially inhalable particulate matter (PM), such as PM 2.5 and PM 10 is proven to increase the morbidity and mortality of the population [2,3]. Meanwhile, the outbreak of COVID-19 has changed the way people live, work, socialise and integrate with urban spaces [4]. This disease has led to a rethinking of the relationship between the urban eco-environment and public health. China is experiencing a transformation of its growing focus on improving its eco-environment, which includes natural elements and processes, as well as anthropogenic activity. Exploring the relationship between urban vibrancy and eco-environment is critical to address major public health emergencies and ensure sustainable urban development. 

Scholars define urban vibrancy from various perspectives. Jacobs [5] first proposed this concept as a condition where “liveness and variety attract more liveness” and emphasised that it primarily refers to humans and their activities in streets. Gehl described urban vibrancy as the vitality of a public place, regardless of the number of occupants [6]. Moreover, measurement methods of urban vibrancy are developing based on Jacobs’s six conditions for urban vitality, namely, land use mix, density, block size, building age and accessibility [5]. Thereafter, extended quantitative measures of the built environment such as urban morphology [7], urban form or design indicators and some “vibrant” characteristics, including walking [8], social [9] and economic [10] activities, have emerged. In the era of big data, geo-referenced datasets including traditional location-based, social network and human mobility data sets, are applied to quantify vibrancy [11].

In addition, scholars have focused on the impact of urban vibrancy on the urban eco-environment for a long time, and empirical studies have shown the intrinsic relationship between urban vibrancy and the urban environment. Based on the exploration of urban vibrancy measurement, research has revealed that urban vibrancy is closely related to the urban environment through the analysis of the association between urban form metrics and intensity of human activities [12]. For instance, building density can change land temperature by changing the speed of near surface airflows [13]. High intensity of human activity, including high intensity of entertainment and recreation, public facilities and other factors, closely correlate with high surface temperature [14]. It is also acknowledged that urban forms including building morphology, transportation system, public infrastructure, ecological infrastructure, and human activity affect urban thermal environments [15]. The close relationship between the hotspots of land surface temperature and the distribution of built-up land has been identified in Zhao et al.’s study [16]. It is also revealed that regional temperature difference can be distinguished by land cover and building surface fraction [17]. Meanwhile, road density has a positive effect on land surface temperature in which high accessibility areas would be hotter than areas with low road density [18]. Various types of land use have different effects on air quality. Air pollution can be caused by industrial and residential activities on land, as well as external transportation, and the impact of industrial air pollution shows spatial variation [19]. It is found that a high degree of urban aggregation is associated with poor air quality in northern China, whereas an opposite association of urban aggregation and air quality has been identified in southern China [20].

In response to the need to cope with public health emergencies, concern about urban vibrancy is increasing in developing countries, such as China, along with the growing focus on environmental protection in the context of global warming and eco-environment deterioration. The impact of urban vibrancy on environmental protection is extensively explored in a quantitative manner. During the COVID-19 pandemic, resilient open spaces, equitable urban communities and quality medical service, which enhance the living conditions of residents and revitalise their communities, have attracted more attention than before [21,22,23]. These factors are closely related to urban vibrancy and the eco-environment. Therefore, analysing the effects of urban vibrancy on the environment can help communities to respond to COVID-19 in targeted ways and improve human wellbeing and public health during the pandemic. Our study aims to fill the gaps in two aspects. First, this study aims to design comprehensive metrics to characterise urban vibrancy with synthesised and geo-referenced mobile and social network data. A single metric or data set is insufficient to perform a systematic reflection on the characteristics of urban vibrancy in modern cities. Density, accessibility, land use and human activities are expected to be included with a powerful big dataset. Second, the relationship between urban vibrancy and the eco-environment remains unclear, although research assumptions claim an impact on the terrestrial, atmospheric and aquatic environments. Remote sensing and other advanced monitoring techniques have enabled a fine-scale assessment on the eco-environment. Therefore, quantitative investigations have been conducted to explore the effects of urban vibrancy on environmental indicators, which can serve as a reference in formulating strategies for sustainable urban development, public health security and human wellbeing promotion. The rest of this paper is structured as follows. Section 2 introduces the study area and methodologies. Section 3 describes the results. Section 4 and Section 5 present the discussion and conclusion, respectively.

## 2. Materials and Methods

### 2.1. Study Area and Data Source

Wuhan was selected as the case study area because of its strategic position in central China, as well as its experience during the COVID-19 pandemic. It is the capital city of Hubei Province and has been regarded as the core city in the Yangtze River Economic Belt (Figure 1). In 2020, the permanent population of Wuhan reached 12.32 million, and the total GDP had grown to 1561.61 billion. The National Development and Reform Commission has designated Wuhan as the country’s National Economic Centre, High-tech Innovation Centre, Trade and Logistic Centre and International Exchange Centre. Wuhan is also the transportation hub in central China and has a bus rapid system through the downtown and interlocking highways, roads and bridges. Wuhan had 10,170 bus stations and five subway lines in 2017 (98.48% of subway stations have bus stations within 1 km; therefore, subway stations are not used in an accessibility calculation system). Wuhan also possesses abundant natural resources with considerable water areas, such as rivers and lakes, which make eco-environmental protection a critical issue in the city [24]. The rapid urban expansion has not only led to environmental degradation and widespread urban decline but also put forward higher requirements for residents’ health and wellbeing in the past decades. Thus, the urban renewal plans have been promoted to improve vitality in the urban space, thereby realizing the efficiency of resource use, enhancing human health and wellbeing, and supporting sustainable development. 

Considering the evidence from empirical studies and data availability, the spatial scales of community neighbourhood are regarded as the basic observation units. In this study, we examined 1074 community neighbourhoods as research units. Due to the outbreak of COVID-19, data from 2019 to 2021 are all affected, and these datasets do not easily reflect the true urban vitality of Wuhan. All data which we use to analyse the urban vibrancy of Wuhan are from around 2017. The datasets we used to measure urban vibrancy include those on population, points of interest (POI), road networks, land use and mobile phone signalling. Population data are obtained from Wuhan statistical yearbooks published by the local government (http://tjj.wuhan.gov.cn/, accessed on 1 September 2018). POI data including geographic points of schools, shops, hospitals and bus stops were collected from the Baidu website (http://map.baidu.com/, accessed on 1 September 2018), one of the largest Chinese search engines. Land use data were obtained from the National Geomatics Centre of China. Land use classification is aggregated into six categories: cultivated land, forest land, grassland, water area, construction land and unused land. Mobile phone signalling data included social media check-ins and mobile phone positioning records. The check-in data were obtained from Sina Weibo in 2017. Mobile phone positioning records were provided by a mobile communication service company in Wuhan. Weekend data (from 7 to 8 October 2017) and workday data (9 October 2017) containing 2441 trajectories were used to calculate urban vibrancy.

### 2.2. Urban Vibrancy Indicators and Assessment

In this study, urban vibrancy is assessed through the DALDH (density, accessibility, liveability, diversity, and human activity) model. Dimensions, indicators, data sources and corresponding years involved in this model are described in detail in Table 1. In accordance with our previous studies comparing urban vitality in Wuhan and Chicago [25] and to accommodate the trend of emphasising human activity and using “big” and “dynamic” data, density includes the indicators of population, road and building density, the density of mobile users and floor area ratio (total floor area above ground divided by the net land area, FAR). The density of mobile users reflects the spatial distribution of flowing population, and FAR embodies the 3D characteristics of buildings, thereby characterising the dynamic and multi-dimensional features in density. Accessibility is measured by the distance of urban facilities, which reflects the convenience and achievement of urban function, and liveability is measured by the number of urban facilities which exhibit the capacity to satisfy all-around living requirements. Diversity measures mixed land. Urban vibrancy is directly related to people’s daily lives and affects human health and wellbeing. We have embraced the dimension of human activity, which is estimated using mobile and social network data [26,27,28]. In particular, the measurement of human activity is conducted by calculating the inflow and outflow numbers and density of Weibo users [29,30]. Human activity is measured by the inflow, outflow and total flow (sum of inflow and outflow) numbers and the number of social network platform check-ins. These data contain information on human activity, which can better capture details of people’s daily life and accurately measure the urban vitality [31].

The values of urban vibrancy and different dimensions are calculated as follows:

First, we obtained the data needed for the calculation from different sources and then normalized the values of the different indicators to unify the range from 0 to 1.

Second, we used the entropy weight method to determine the index weight. The entropy weight method is an assignment method to determine the weight of each index by the size of information entropy. The index weight value determined by this method reduces the human subjective interference [32,33]. $W_{DENV},\;W_{AV},\;W_{LV},W_{DIVV},\;W_{HAD}$ are the corresponding weights of $DENV_{\left(density\right)}$, $AV_{\left(Accessibility\right)}$, $LV_{\left(Livability\right)}$, $DIVV_{\left(Diversity\right)}$ and $HAD_{\left(Human\;activity\right)}$, respectively.$\;W_{deni}$ is the weight of each index in the dimension of density. $W_{acci}$ is the weight of each index in the dimension of accessibility.$\;W_{livi}$ is the weight of each index in the dimension of accessibility. $W_{hadi}$ is the weight of each index in the dimension of human activity.

Third, we calculated the values of urban vibrancy in different dimensions based on the weights and their normalized values as follows (Equations (1)–(6)).
(1)$$\begin{array}{ll} VRC_{i} & =W_{DENV}*DENV_{i}+W_{AV}*AV_{i}+W_{LV}*LV_{i}+W_{DIVV}*DIVV_{i}+W_{HAD}*HAD_{i}VRC_{i}\\ & =W_{DENV}*DENV_{i}+W_{AV}*AV_{i}+W_{LV}*LV_{i}+W_{DIVV}*DIVV_{i}+W_{HAD}*HAD_{i} \end{array}$$
(2)$$\begin{array}{ll} DENV_{\left(density\right)} & =W_{den1}*Den_{pop}+W_{den2}*Den_{bud}+W_{den3}*Den_{mob}+W_{den4}*Den_{flr}+W_{den5}*Den_{rod}DENV_{\left(density\right)}\\ & =W_{den1}*Den_{pop}+W_{den2}*Den_{bud}+W_{den3}*Den_{mob}+W_{den4}*Den_{flr}+W_{den5}*Den_{rod} \end{array}$$
(3)$$\begin{array}{ll} AV_{\left(Accessibility\right)} & =W_{acc1}*Dist_{sch}+W_{acc2}*Dist_{hosp}+W_{acc3}*Dist_{shp}+W_{acc4}*Dist_{bus}AV_{\left(Accessibility\right)}\\ & =W_{acc1}*Dist_{sch}+W_{acc2}*Dist_{hosp}+W_{acc3}*Dist_{shp}+W_{acc4}*Dist_{bus} \end{array}$$
(4)$$\begin{array}{ll} LV_{\left(Livability\right)} & =W_{liv1}*Num_{bks}+W_{liv2}*Num_{fd}+W_{liv3}*Num_{lf}+W_{liv4}*Num_{lsr}LV_{\left(Livability\right)}\\ & =W_{liv1}*Num_{bks}+W_{liv2}*Num_{fd}+W_{liv3}*Num_{lf}+W_{liv4}*Num_{lsr} \end{array}$$
(5)$$DIVV_{\left(Diversity\right)}=-\sum_{i=1}^{n}p_{i}lnp_{i}DIVV_{\left(Diversity\right)}=-\sum_{i=1}^{n}p_{i}lnp_{i}$$
(6)$$\begin{array}{ll} HAD_{\left(Human\;activity\right)} & =W_{had1}*Infl+W_{had2}*Ourfl+W_{had3}*Tfl+W_{had4}*WCinHAD_{\left(Human\;activity\right)}\\ & =W_{had1}*Infl+W_{had2}*Ourfl+W_{had3}*Tfl+W_{had4}*WCin \end{array}$$
where VRC refers to the urban vibrancy value; $DENV_{density}$ is the density value which includes population density ($Den_{pop}$), building density ($Den_{bud}$), density of mobile users ($Den_{mob}$), floor ratio area ($Den_{flr}$), and road density ($Den_{rod}$); $AV_{Accessibility}$ is the accessibility value that is calculated using the indices of distance to school ($Dist_{sch}$), hospital ($Dist_{hosp}$), shop ($Dist_{shp}$), and bus stop ($Dist_{bus}$); $LV_{Livability}$ is the liveability value that is calculated based on the number of banks ($Num_{bks}$), food service sites ($Num_{fd}$), life service sites ($Num_{lf}$), and leisure sites ($Num_{lsr}$); $DIVV_{diversity}$ is the diversity value measured using the Shannon diversity index ($p_{i}$ is the percentage of *i*th land use parcel, n is the number of parcels); and $HAD_{Human\;activity}$ is the human activity dimension using the indices of inflow and outflow number (Infl,Outfl), total number of mobile flows (Tfl ), and geo-referenced social network users on Weibo (WCin).

Finally, to combine all the values in the DALDH framework, we employed the spatial technique for order preference by similarity to ideal solution (Technique for Order Preference by Similarity to an Ideal Solution (TOPSIS) method to produce the ultimate urban vibrancy value. Hwang and Yoon [34] proposed the traditional TOPSIS to identify the optimal solutions from a finite set of alternatives using multiple criteria. Spatial TOPSIS modifies the traditional TOPSIS method by using the Euclidean distances as the weight to calculate the gap between the actual and optimal values for each observation [25]. This modification involves the incorporation of spatial interaction in the assessment because spatial influence in urban space is a factor that cannot be disregarded. The neighbouring communities are capable of influencing the local communities because of the critical nature of the inner urban network. Hence, the principle of the calculation in our study is that we take the distance to neighbouring communities into account in additions to their attribute values when measuring the values of gaps among the observations. This distance is applied in traditional TOPSIS as the components of spatial weights to form spatial-TOPSIS. Here are the main calculation step for spatial-TOPSIS ((Equation (7)) and the computing method of spatial weights (Equation (8)):(7)$$G_{ij}=\sqrt{\overset{m}{\underset{j}{\sum }}w_{ij}*{\left(f_{ij}-f_{imax}\right)}^{2}}$$
(8)$$w_{ij}=\frac{DIST_{ij}}{\sum_{j=1}^{m}DIST_{j}}$$
where $G_{ij}$ is the gap value of the *j*th observation for the *i*th candidate which indicates the dimensions of density, accessibility, liveability, diversity and human activity. $f_{ij}$ is the *j*th observation in the *i*th dimension, $f_{imax}$ is the highest value in the *i*th dimension. $DIST_{ij}$ is the spatial distance of *j*th observation to the ideal solution in the *i*th dimension, which is the maximum value in the *i*th dimension. In this paper, we take the highest value as the ideal solution as the ideal solution in each dimension. $DIST_{j}$ is the sum of the distances from the *j*th observation to all other observations in the *i*th dimension. $w_{Dij}$ is the spatial distance weight of *j*th observation in the *i*th dimension. Then, we ranked the communities according to the value of $G_{ij}$. 

### 2.3. Retrieval of Eco-Environmental Indicators

In the context of global warming and increasing concern about air pollution in China, we applied two indicators, namely, PM 2.5 concentrations and temperature, to reflect the eco-environment quality with consideration of data accessibility and urban sustainability issues [35,36]. In terms of temperatures, because abnormal temperature was observed from 2017 to 2019 and data acquisition restrictions were encountered due to the COVID-19 pandemic, we used the remote sensing inversion model to retrieve the surface temperature from the thermal infrared sensor on board the Landsat 8 with spatial resolution of 30 m in 2016. The practical split-window algorithm with consideration of atmospheric water vapor was applied to retrieve land surface temperature (LST) using the professional software development of Ren et al. [37,38]. We used the global annual PM 2.5 grids from Moderate-resolution Imaging Spectroradiometer (MODIS) provided by the socioeconomic data and applications centre in 2016. The raster grids present the values of PM 2.5 concentrations with cell resolution of 0.01° (https://sedac.ciesin.columbia.edu/data, accessed on 1 September 2018). The retrieved LST and PM 2.5 data set are overlaid with urban communities in Wuhan to acquire the mean land surface temperature and PM 2.5 concentrations in each urban community. Given the substantial differences in the areas of urban communities, the treatment of the values for each community conforms to different principles. When the cell size of the LST or PM 2.5 grids is larger than the community area, the value of the cell is the value in the community. When the cell size of the LST or PM 2.5 grids is smaller than the community area, the average values of all the covered cells are used as the values in the community.

### 2.4. Effects of Urban Vibrancy on Eco-Environment

We assumed that there are spatial interactions among eco-environmental and DALDH indicators according to the empirical studies [39,40]. As a result, the spatial econometric model was applied to explore the relationship between urban eco-environmental indicators and DALDL indicators. The dependent variables are land surface temperature and PM 2.5. The explanatory variables are density, accessibility, liveability, diversity, and human activity in both situations. Thereafter, we use Moran’s I index and Lagrange multiplier (LM) statistics to diagnose the spatial autocorrelation and spatial econometric model selection. Moran’s index is a measure of spatial autocorrelation and has been widely used in various studies to examine the existence of correlations in a signal among nearby locations in space [41,42]. LM is an indicator of spatial dependence that was first derived in Anselin’s study [43]. Together with Moran’s I, this statistic has also been embedded in spatial econometric packages in various software to identify spatial dependence and provide guidance for determining the existence of spatial autocorrelation in lags or errors [42]. The current study uses LMerror and LMsar to test the spatial autocorrelation in the residuals after the ordinary least squares (OLS) regression to check the significance level for the determination of suitable spatial econometric models. The basic forms of the spatial lag model (Equation (9)) and spatial error model (Equation (10)) are as follows: (9)$$\mathrm{Temp}/\mathrm{PM}\;2.5_{i}={\alpha w}_{ij}\mathrm{Temp}/\mathrm{PM}\;2.5_{j}+{M*X}_{\mathrm{DALDH}}{}_{i}+\varepsilon$$
(10)$$\mathrm{Temp}/\mathrm{PM}\;2.5_{i}={\beta X}_{\mathrm{DALDH}}{}_{i}+\lambda \mathrm{Wu}+\varepsilon$$
where *i* and *j* refer to the different areas, $\mathrm{Temp}/\mathrm{PM}\;2.5_{i}$ is the value of land surface temperature or PM 2.5 in the *i*th urban community. $X_{\mathrm{DALDH}}{}_{i}$ is the DALDH values in the *i*th urban community. $w_{ij}$ is the N × N order spatial weights matrix of spatially lagged response variables ($\mathrm{Temp}/\mathrm{PM}\;2.5_{j}$). α denotes spatial autoregressive coefficient, and ε is the independently distributed errors. u is spatially lagged errors. W is the N × N order spatial weights matrix of the spatially lagged error (u). β is the regression coefficient of independent variable.  λ  represents the spatial error regression coefficient.

Various types of interaction effects occur between the spatial lag model (SLM) and spatial error model (SEM). The first model contains endogenous interaction effects among the dependent variable (Temp/PM 2.5) and the other interaction effects among the error terms (λ). Generally, interaction effects can describe why a dependency exists between an observation in a specific location and observations at other locations [12]. Endogenous interaction effects mean that the dependent variable of a particular unit A depends on the dependent variable of other units such as unit B. The interaction effects among the error terms are matched with a situation where determinants of the omitted dependent variable in the model are spatially autocorrelated, or with a situation where the unobservable blows are associated with a spatial pattern [44]. According to LMerror and LMsar, SLM is taken in Qiaokou, while SEM is suitable for other locations in the spatial regression for PM 2.5. For land surface temperature, OLS is taken in Jiangan and Qingshan, while SEM is suitable for other locations.

## 3. Results

### 3.1. Urban Vibrancy Assessment

The results of the urban vibrancy assessment have shown great differences in different dimensions. Table 2 shows the maximum value, minimum value, mean value, deviation in the DALDH dimensions, and integrated urban vibrancy. Figure 2 illustrates the spatial distribution of the urban vibrancy values. In general, accessibility has the highest mean value and diversity has the largest deviation. Integrated urban vibrancy has a mean value of 0.1085 and deviation of 0.002. The spatial distribution of density, diversity, and human activity shows a similar pattern, where the highest values are scattered in the southeastern and middle-eastern areas of the Hongshan district. By contrast, accessibility and liveability exhibit high degrees of similarity in spatial distribution with clustering trend of high values along the Yangtze River and the horizontal axis in the eastern area. Mobile data record human mobility, and we extracted individual flows in three days to analyse the inflow and outflow in each urban community (Figure 3). To eliminate the “fault” and “problem” data and guarantee data consistency, we selected 2441 trajectories in analysing the population flow. In general, we observed no substantial difference in the spatial distribution of inflow and outflow, thereby indicating that the phenomenon of “high-in and low-out” or “high-out and low-in” hardly appears. The urban communities with the highest human mobility are in the Hongshan district in the eastern area of Wuhan, where several prestigious universities (e.g., Wuhan University and Huazhong University of Science and Technology) and scenic parks (e.g., eco-tourism scenic spots in East Lake and a botanical garden) are located. The lowest “mobilised” communities are along the Yangtze River, where piers, urban villages (a dualistic phenomenon showing the coexistence of city and village caused by the fast urban expansion in the urbanisation process [45]), and derelict factories are located, such as in Xiwan Pier, Rocket Village, and Qingshan Shipyard, where land use function is generally one-fold, and the frequency of the population flow has a low level. Thus, urban communities with the highest urban vibrancy are spread in the outer rings and clustered in the western area along the Yangtze River.

The communities with the highest urban vibrancy values are the TongFu and Xiaojia urban communities. They are old towns in the Jiangan district and are located in a bustling commercial district near the famous Jianghan Pedestrian Street. These communities have extensive facilities, including those related to housekeeping, maintenance, food, and medical treatment, and the population size and flow are evident. The third one is Guanshan Village in the Hongshan district surrounded by several prestigious universities with a lively population flow. In all these communities with high urban vibrancy, old villages and new buildings are found with a series of urban regeneration projects and a transportation facility that has substantially improved in recent years.

### 3.2. The Spatial Distribution of PM 2.5 and Land Surface Temperature

The spatial distributions of PM 2.5 and surface temperature were then analysed. The spatial distribution of PM 2.5 presents a progressive decrease pattern from the northwestern to the southeastern area, with the highest and lowest values of 53.6 and 48.8 mg/m^3^, respectively (Figure 4 and Figure 5). The mean values of PM 2.5 do not make a substantial difference with Jiangan, Jianghan and Qiaokou in the western area along the Yangtze River, having slightly high values (Figure 6, the first number is the mean value of PM 2.5, and the second number is the stand deviation). The Hongshan urban district has the lowest PM 2.5 values that are generally attributed to the improvement effect from East Lake and abundant forest cover. Industrial development and several development zones, such as the Pan Longcheng Economic Development Zone(Wuhan, China), Wuhan Economic and Technological Development Zone(Wuhan, China), and Changfu Xin Cheng Economic Development Zone(Wuhan, China), contribute to the severe air pollution in the northwestern area. The central urban communities, where railway stations and tunnels across the Yangtze River are located, have also suffered from air pollution. 

High PM 2.5 values are generally attributed to industrial development and infrastructure construction in urban areas. The World Health Organization (WHO) has issued air quality guidelines in which the standards have been formulated (the annual averages of air quality guideline level is 5 μg/m^3^ and interim targets 1 to 4 are 35, 25, 15 and 10 μg/m^3^. The 24 h average of air quality guideline level is 15 μg/m^3^ and interim targets 1 to 4 are 75, 50, 37.5, 25 μg/m^3^ [46]). The ambient air quality standards in China indicate that the first-level PM 2.5 concentration limit in area I is the annual average of 15 μg/m^3^ and the 24 h average of 35 μg/m^3^; the second-level PM 2.5 concentration limit in area II is the annual average of 35 μg/m^3^ and the 24 h average of 75 μg/m^3^ [47]. Compared with the international standards, China meets the relaxed standards set by the WHO. Although the overall PM 2.5 in Wuhan is in line with the Chinese concentration limit, it is in interim targets 1 and 2 in the international standards, which are still far from the air quality guideline level. Among them, urban districts with the highest PM 2.5 are located along the outer ring in the northwestern area. The Yuhualing and Taizihu urban communities are transformed from the regeneration of urban villages in the Jiangan urban district. This area has massive real estate development and infrastructure construction, thereby resulting in high PM 2.5 values in recent years. Urban districts with the highest PM 2.5 are located along the outer ring in the northwestern area. The Yuhualing and Taizihu urban communities are transformed from urban village renovation along the third loop in the Jiangan urban district. This area has massive real estate development and infrastructure construction, thereby resulting in high PM 2.5 values in recent years.

Similarly, the spatial distribution of surface temperature was explored. The surface temperature in Wuhan has apparent spatial heterogeneity and the temperature is higher in the western area than that in the eastern area along the Yangtze River (Figure 7). Figure 8 shows land surface temperature of different districts (the first number is the mean value, and the second number is the stand deviation). The highest temperature occurs along the Yangtze River and Han River at 56 °C, whereas rivers and lakes show a low temperature of 22 °C.

To reveal the differences among communities, we made a Boston Consulting Group (BCG) matrix using two variables (Figure 9): land surface temperature (LST) and PM 2.5. As a whole, the LST of the communities in the study area is generally high, with few communities below 30 °C and most at around 37.5 °C (mean land surface temperature of 37.64 °C). In addition, the PM 2.5 levels in the study area communities are all at a high level (average PM 2.5 of 49.91 ug/m^3^), and almost all communities are in a lightly polluted state according to the PM 2.5 detection standard (light pollution range is 75–115 ug/m^3^). Further exploration reveals that more points exist in the graph with lower PM 2.5 level and higher LST or higher PM 2.5 levels and higher LST. Furthermore, the number of communities with lower LST and higher PM 2.5 level is less. In general, the communities in the study area differed slightly from each other in terms of PM 2.5 levels and LST. 

### 3.3. The Spatial Regression Analysis of PM 2.5 or LST vs. Urban Vibrancy Dimensions

Table 3 shows the results of the spatial regression for PM 2.5 in the different districts. It is noted that a number of pixels indicate missing data of PM 2.5 because of influence by information sources at coarser resolutions. We overlaid the PM 2.5 data with the research area and 346 urban communities were used for modelling, which accounted for almost one-third of the total cells. We analysed the contributions of urban vibrancy to air pollution in seven urban districts, namely, Jiangan, Jianghan, Wuchang, Hongshan, Qiaokou, Qingshan, and Hanyang. Spatial autocorrelation was diagnosed in all the urban districts, and we identified that this spatial influence has high probability to appear in the error terms in nearly all urban districts except Qiaokou. The greatest power of spatial influence in error term is in Qingshan, where Wuhan Iron and Steel Corporation is located. This district also has the highest R2, and urban vibrancy was capable of explaining 84.45% of the variance in PM 2.5. In the Qingshan district, density, accessibility and diversity are positively correlated with PM 2.5, with accessibility having the largest correlation coefficient. By contrast, human activity and PM 2.5 have a negative relationship. Hongshan is the largest urban district in the city centre, and the spatial influence is also of high level with a coefficient of 0.79. However, the only significant dimension is liveability, with a positive correlation coefficient of 7.14. Wuchang, the other urban district on the eastern side along the Yangtze River, has the only significant spatial influence factor with all the dimensions of urban vibrancy insignificantly correlated. On the western side along the Yangtze River, Jianghan has the highest coefficient in spatial influence and highest R2 with a value of 0.76. Density, accessibility and diversity are negatively correlated, and human activity is positively correlated with PM 2.5. Accessibility has the most powerful negative influence on air pollution, which is contrary to the situation in Qingshan. Density is the only significant factor with a negative influence on PM 2.5 and the coefficient of spatial influence is 0.71 in Jiangan. In Hanyang, human activity has a significant effect on air pollution with a correlation coefficient of 3.16, and accessibility and diversity are negatively correlated with urban vibrancy. Qiaokou is the only urban district with spatial auto-correlation appearing in a spatial lag with a coefficient of 0.0161. Accessibility and diversity are two identified factors with a significance level of 0.05, and the previous one shows negative influence, whereas the latter is positive. Urban vibrancy has been justified to have a relationship with air pollution with various levels of contributions in the different urban districts. 

Spatial autocorrelation was also identified to be significant in exploring the relationship between urban vibrancy dimensions and LST in the majority of the urban districts (except in Jiangan and Qingshan), with the spatial influence being likely to appear in the error term (Table 4). Hanyang has the highest coefficient of spatial influence and R2. Accessibility is the only significant factor in explaining the variance in LST. The coefficients of the spatial error term are similar in Jianghan and Hongshan. In Jianghan, all dimensions have shown significant influences except diversity, with liveability and human activity being negative. In Hongshan, accessibility and diversity are the significant factors with the former being positive. The contributions of the spatial influences are weak in Wuchang and Qiaokou. In Wuchang, the only positive significant factor is accessibility, whereas density, diversity, and human activity show a negative impact, with density being more powerful. In Qiaokou, diversity is the only significant factor with a correlation coefficient of −3.63. No significant spatial autocorrelation was observed in Jiangan and Qingshan. All factors are significantly correlated with LST in Jiangan except diversity. By contrast, only diversity is negatively correlated with LSI in Qingshan. 

The correlation between different indicators of urban vibrancy and LST or PM 2.5 varies from one district to another. Both positive and negative correlations exist. Based on the results of the spatial regression for PM 2.5 and LST, the results of the spatial regression for PM 2.5 are better, except in the Wuchang, Qiaokou and Hanyang districts, where R² is above 0.5. In addition to Hanyang district, the R² of the results of the spatial regression for PM 2.5 in other districts are better than the results of the spatial regression for LST. From the results of the spatial regression for PM 2.5, Qingshan district has the best fitting effect where R² is 0.8445. According to the results of the spatial regression for LST, Hanyang district has the best fitting effect, but its R² is only 0.5380, and the R² of other districts are in the range of 0.3 and 0.4.

## 4. Discussion

The integrated assessment on urban vibrancy using big and dynamic data and the exploration of its relationship with the urban eco-environment are imperative for urban revitalization, sustainable urban development, public health and human wellbeing. The primary contributions of our study are in the integrated assessment of urban vibrancy with the extended DALDH indicators and the investigation of the effects of urban vibrancy on the urban eco-environment. Conceptually, we assessed urban vibrancy by considering features such as density, diversity and human activity and driving factors such as accessibility and liveability. The assessment of urban vibrancy has been implemented in our previous study in Wuhan and Chicago using the DALD model [25]. We extended our indicators of density to include FAR to manifest the vertical dimension, and the dimension of human activity was embedded, thereby conforming to the mainstream in this domain. In this dimension, mobile and social network data were used to indicate the “popularity” degree in urban community, which is a critical indicator of urban vibrancy in Jacob’s theory [5]. Contemporarily, the urban village was revealed to be of low-density value, whereas a relatively new urban community (characterised by marketized real estate with various facilities and stakeholders [48]) has high-density values. These results are similar to the situation in the dimensions of accessibility and liveability. A high degree of mixed land use often occurs in urban communities in the outer ring and with a large area, such as the Wudong community. Substantial population flow and social network are apparent in urban villages in the Hongshan district, such as Qingling Village. Urban communities with the highest level of urban vibrancy generally have small areas and are along the Yangtze River on the west side, which is the commercial centre or lake area on the east side. This condition revealed urban regeneration functions in Wuhan, whereas the closeness of interpersonal relationships is higher in the old communities than in the newly built ones. Consequently, efforts should be exerted to create dynamic and networked urban communities at the current stage to realise vibrant urban development. 

Remote sensing techniques have provided immense opportunities for the quantification of the eco-environment at a fine scale [12]. We selected PM 2.5 and LST as indicators to retrieve data from remotely sensed images because air pollution and urban heat island effect are primary environmental concerns in Wuhan. Significant spatial autocorrelation and the relationship between urban vibrancy and eco-environment were observed regardless of PM 2.5 or LST. Although urban vibrancy is incapable of explaining the variance in PM 2.5 or LST in the entire research area, it is predictable that modelling their relationship in each urban district may produce different results. Spatial modelling techniques are applied in this process because of the spatial autocorrelation phenomenon, and they improved the model fitting considerably compared with the traditional regression with ordinary least squares method [49]. For the modelling of PM 2.5, spatial influence is the most powerful in Qingshan, where air pollution is most severe because of the steel and iron sectors development. This spatial effect is considerably evident in the Hongshan district with respect to LST and where East Lake and the East Lake Hi-tech Economic Zone are located. It is also the urban district with the largest land area where most prestigious universities are clustered. Thus, the industrial, knowledge and environmental spatial spillover effect has a high probability to appear. Regularity in terms of the effects of density, accessibility, liveability, diversity, and human activity on PM 2.5 or LST is difficult to determine. For example, negative relationships were identified in Jianghan, but positive ones were observed in Qingshan with respect to density and PM 2.5. However, a positive correlation coefficient was observed between density and LST in Jianghan but was negative in the neighbouring district of Jiangan. For the exploration of the effects of urban vibrancy on LST, diversity and human activity showed significant negative impact in the majority of the urban districts. This result is generally attributed to the cooling function of the water area and vegetation cover and urban heat island effect caused by population clustering. Spatial influence and spatial heterogeneity in the effects of urban vibrancy on the urban eco-environment is beneficial to guide the elaborated urban planning in infrastructure construction, land use, population control and other related policies to improve air quality and sustain the liveable surface temperature in each urban district and the entire city centre [50].

This study shows a spatial autocorrelation and relationship between urban vibrancy and the eco-environment. Meanwhile, many studies have linked urban eco-environment with public health from different aspects. Furthermore, improving urban vibrancy is one of the goals of urban planning, and it is important to recognise its role in improving the urban eco-environment, enhancing public health and reducing the negative impacts of public health emergencies through scientific urban planning. Among the indicators selected for urban vibrancy in this study, density, population flow and accessibility all reflect the aggregation in the distribution of population, transportation facilities and public service facilities in Wuhan. This overconcentrated urban spatial pattern poses a potential threat in the form of a rapid spread of epidemics. In terms of the diversity of land use, the insufficiency of public space is a hidden danger in reserving emergency sites during an epidemic. Accordingly, urban planning in the post-epidemic era can be improved by strengthening mixed land use, increasing open and green spaces, as well as balancing the distribution of public service facilities and optimised transportation road networks. Researchers suggest that open spaces are crucial for residents’ mental wellbeing and human activity [51,52]. Human wellbeing is also improved by increasing green spaces and their percentage of vegetation cover and size in all aspects [53,54]. Enhancing urban vibrancy is crucial in realising the human-oriented urban planning and intrinsically improving the urban environment. Eco-environment is the key to shaping public health and human wellbeing. Thus, research on urban vibrancy could help decision makers to integrate the eco-environment into urban planning to improve public health and wellbeing by optimising the living space and flow in the post-epidemic era.

## 5. Conclusions

This study assessed urban vibrancy from the DALDH dimensions and explored its effects on the urban eco-environment through the interpretation of PM 2.5 and land surface temperature from remote sensing images. Spatial heterogeneity is observed in the urban vibrancy distribution and its impact on the urban eco-environment. Density, diversity and human activity showed similar spatial patterns with the highest values clustering in the old town and high-tech and university areas, which determine the spatial distribution of the ultimate urban vibrancy. Improved accessibility and liveability are apparent in urban communities in the city centres along the Yangtze River. It is also revealed that severe air pollution and urban heat island effect have been diagnosed in central business districts due to the rapid socio-economic development in the past decades, as identified in other cities in China and around the world. Conversely, the effects of urban vibrancy on urban eco-environment are complex, as different dimensions of urban vibrancy have shown different effects with different magnitude. In our study, it was found that the advancement of urban vibrancy in density, accessibility, liveability and diversity may improve air quality. However, dense population flow brings air pollution, and the spatial balance of population distribution can be a feasible method to reduce this negative effect. Meanwhile, increasing accessibility also raises the surface temperature in most of the districts, and large-scale infrastructure construction generally contributes to this increment. Diversity and human activity appear to have a cooling effect. A high level of mixed land use is suggested to mitigate the global warming effect. In the future, spatial heterogeneity may be applied in urban vibrancy assessment and its effects on the urban eco-environment to achieve sustainable urban development. When urban vibrancy is assessed or optimised, diversity and human activity have to be considered. Meanwhile, urban vibrancy has a close relationship with the urban eco-environment, and their interaction emerges as an important issue in ensuring public health and promoting human wellbeing.

## Figures and Tables

**Figure 1 ijerph-19-03200-f001:**
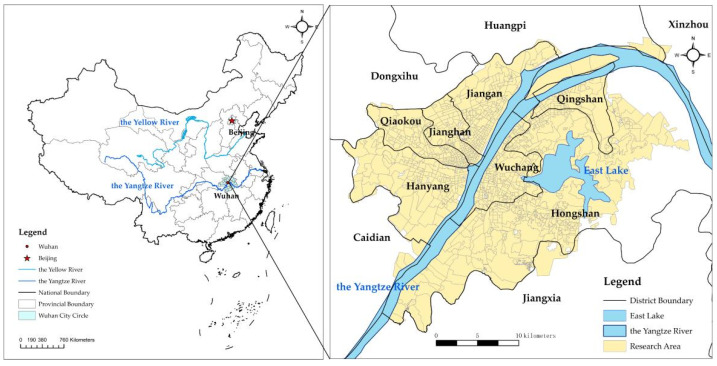
Location of research area (Wuhan).

**Figure 2 ijerph-19-03200-f002:**
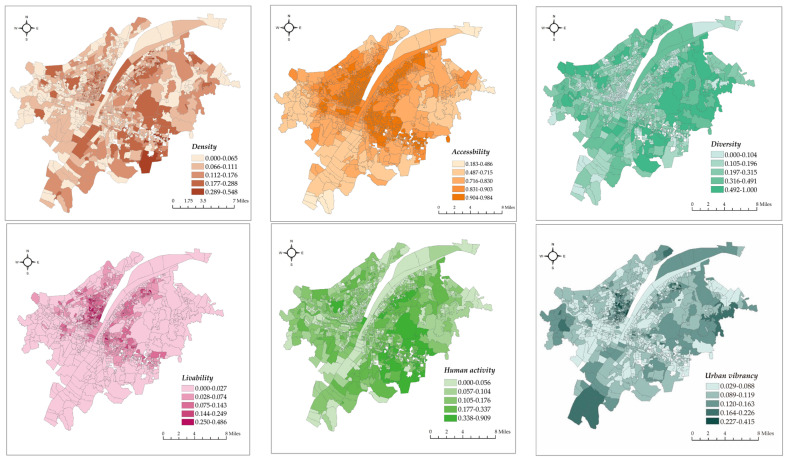
Spatial distribution of the values of urban vibrancy and of its dimensions.

**Figure 3 ijerph-19-03200-f003:**
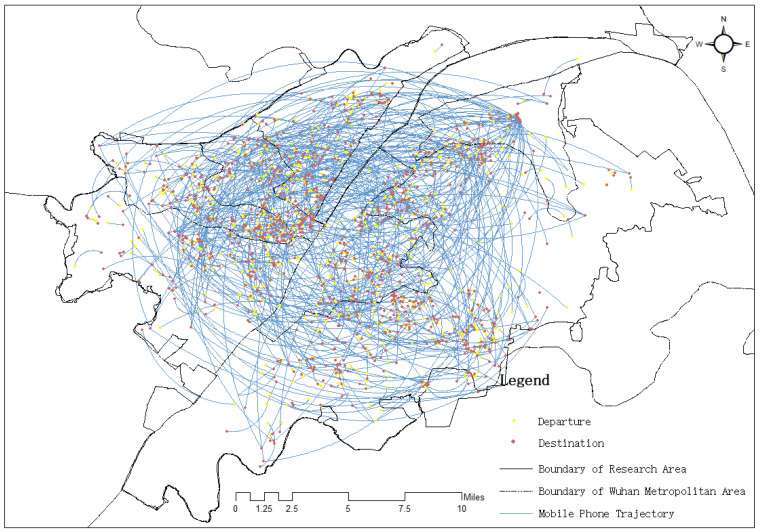
Mobile phone trajectories.

**Figure 4 ijerph-19-03200-f004:**
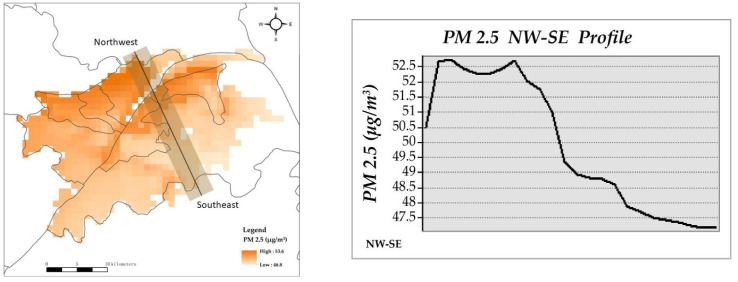
Distribution of PM 2.5 values and PM 2.5 NW–SE profile.

**Figure 5 ijerph-19-03200-f005:**
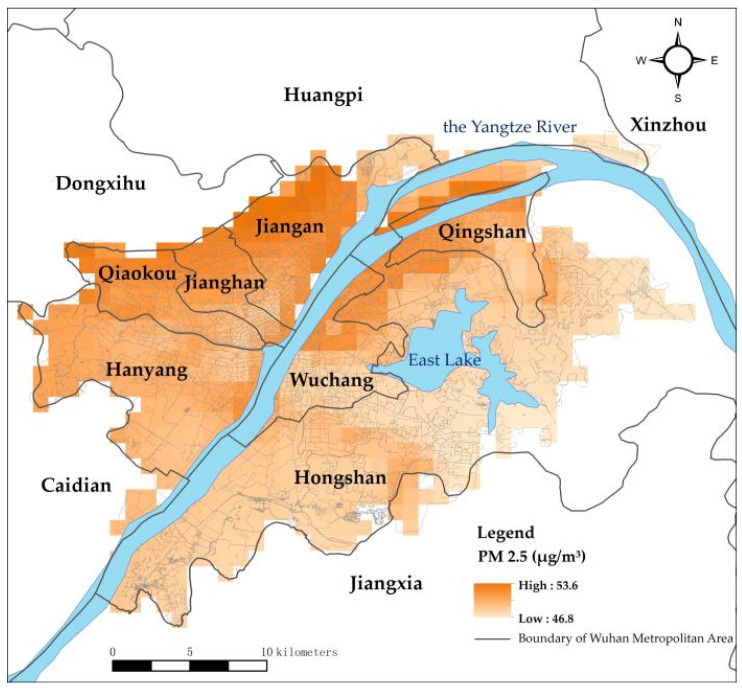
Spatial distribution of PM 2.5.

**Figure 6 ijerph-19-03200-f006:**
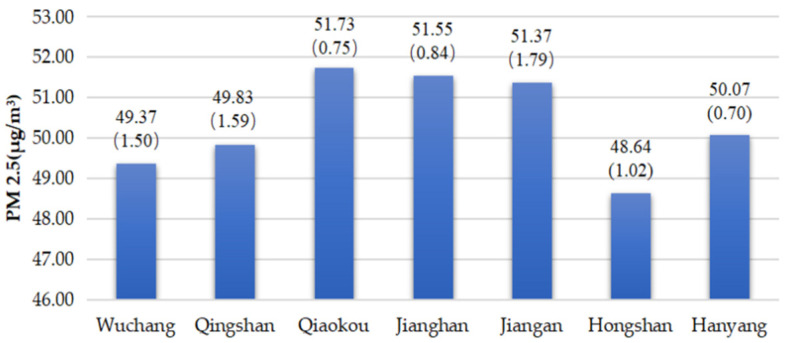
PM 2.5 values of different districts.

**Figure 7 ijerph-19-03200-f007:**
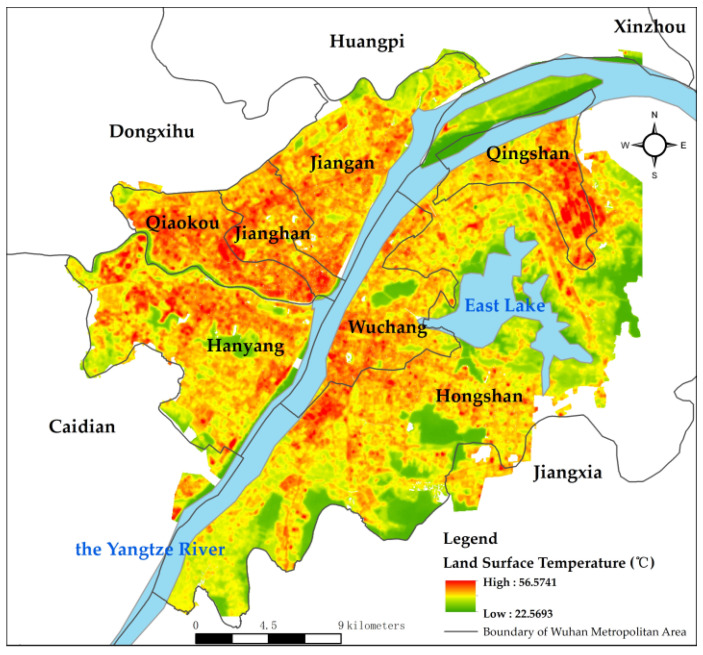
Spatial distribution of LST.

**Figure 8 ijerph-19-03200-f008:**
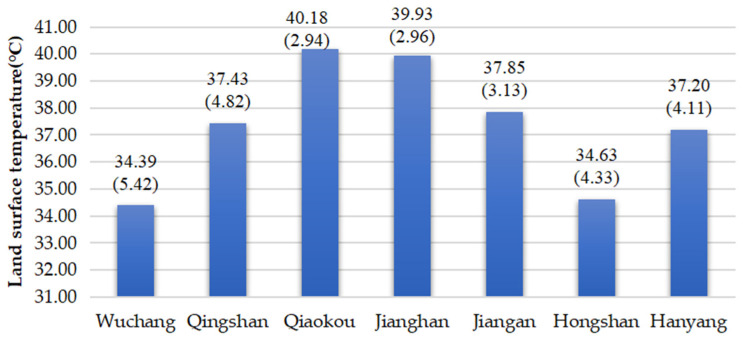
Land surface temperature of different districts.

**Figure 9 ijerph-19-03200-f009:**
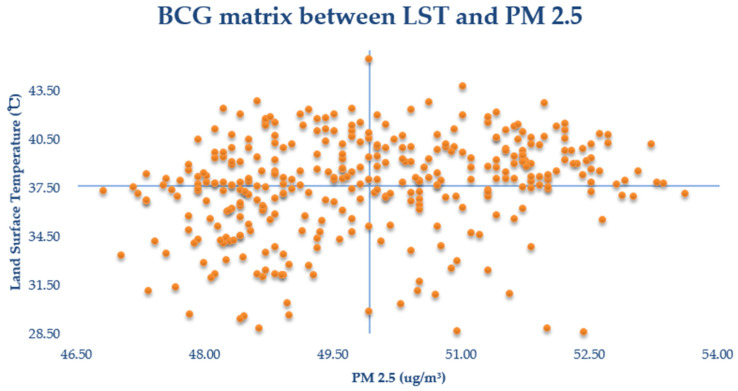
Scatterplot between LST and PM 2.5.

**Table 1 ijerph-19-03200-t001:** Descriptions of DALDH model.

Dimensions	Indicators	Data Source	Year
Density	Population density	Census data set from local government	2017
	Building density	Wuhan Natural Resources and Planning Bureau	2017
	Density of mobile users	Mobile phone GPS positioning requests	2017
	Floor Area Ratio	Wuhan Natural Resources and Planning Bureau	2017
	Road density	Wuhan Natural Resources and Planning Bureau	2017
Accessibility	Distance to school	Big data platform (Baidu API)	2017
	Distance to hospital	Big data platform (Baidu API)	2017
	Distance to shop	Big data platform (Baidu API)	2017
	Distance to bus stop	Big data platform (Baidu API)	2017
Liveability	Number of banks	Big data platform (Baidu API)	2017
	Number of food service sites	Big data platform (Baidu API)	2017
	Number of life service sites	Big data platform (Baidu API)	2017
	Number of leisure sites	Big data platform (Baidu API)	2017
Diversity	Land use diversity	National Geomatics Centre of China	2017
Human activity	Inflow	Mobile phone GPS positioning requests	2017
	Outflow	Mobile phone GPS positioning requests	2017
	Total Flow	Mobile phone GPS positioning requests	2017
	Weibo check-in	Social network platform (Weibo)	2017

Note: Baidu API, Application Programming Interface, a developer’s open data platform; GPS, Global Positioning system.

**Table 2 ijerph-19-03200-t002:** Normalised values in different dimensions.

Dimension	Density	Accessibility	Liveability	Diversity	Human Activity	Urban Vibrancy
Mean	0.0985	0.865	0.0477	0.184	0.0932	0.1085
SD	0.0038	0.0096	0.0043	0.0201	0.0057	0.002

Note: Mean indicates mean value, and SD indicates standard deviation.

**Table 3 ijerph-19-03200-t003:** Results of spatial regression for PM 2.5 in different districts.

	Jiangan	Jianghan	Wuchang	Hongshan	Qiaokou	Qingshan	Hanyang
Observation	50	31	62	154	42	34	63
Density	−5.77 ***	−12.62 ***	0.1563	0.4902	−4.14	2.08 ***	−1.13
Accessibility	0.7992	−14.37 ***	−0.2465	0.3763	−3.59 **	6.27 **	−2.08 *
Liveability	−0.0425	0.5554	−3.26	7.14 ***	4.77	−1.24	3.08
Diversity	0.6657	−1.36 **	−0.1600	0.0696	2.16 **	0.5472 *	−1.23 **
Human activity	1.44	6.10 ***	−0.1149	−0.1318	2.25	−3.62 **	3.16 ***
α	-	-	-	-	0.0161 **	-	-
λ	0.7122 ***	0.7830 ***	0.5963 ***	0.7903 ***	-	0.8571 ***	0.5583 ***
R^2^	0.6730	0.7607	0.4809	0.6991	0.4828	0.8445	0.4700

Note: *, ** and *** indicate 10%, 5% and 1% significance levels, respectively.

**Table 4 ijerph-19-03200-t004:** Results of spatial regression for land surface temperature in different districts.

	Jiangan	Jianghan	Wuchang	Hongshan	Qiaokou	Qingshan	Hanyang
Observation	207	143	206	316	175	83	141
Density	−4.33 **	5.49 *	−6.03 *	1101	0.9157	−1623	−0.8696
Accessibility	13.08 ***	18.09 ***	9.61 **	2464 ***	1.36	2369	11.40 ***
Liveability	2.85 **	−3.90 **	−0.9617	−1815	1.29	−618	3.32
Diversity	0.0769	0.032	−4.57 ***	−1041 **	−3.63 ***	−2144 ***	−0.4684
Human activity	−6.00 ***	−4.37 *	−4.50 **	648.82	−0.3313	2961	−3.82
λ		0.5097 ***	0.3247 ***	0.5107 ***	0.3027 ***		0.6541 ***
R^2^	0.3511	0.3746	0.3316	0.4082	0.2086	0.2161	0.5380

Note: *, ** and *** indicate 10%, 5% and 1% significance levels, respectively.

## Data Availability

The data that support the findings of this study are available from the corresponding author upon reasonable request.

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
