# Peer review of "Effects of Urban Vibrancy on an Urban Eco-Environment: Case Study on Wuhan City"

_ijerph, 2022, doi:10.3390/ijerph19063200_

Round 1
Reviewer 1 Report
This is a decent paper on the topic of the relationship between urban vibrancy and urban eco-environment at the city scale. Spatial modeling was performed to investigate the influence of urban vibrancy on air pollution and temperature with inverted and extracted remote sensing data in the paper and meaningful conclusions were reached. Although the relationship between urban vibrancy and LST or PM 2.5 is of low level in the entire research area, different dimensions in urban vibrancy were found to have an impact on the eco-environment with different magnitudes in different urban districts. This result provides valuable references for the combination of livelihood issues and spatial urban planning. The paper delves deeper than the previous published works and it is well-organized. However, it should be noted that there is also some space of improvement for this paper. Overall, it is recommended for publication after responding the specific issues below.
(1) I don’t think the urban vibrancy indicators and urban eco-environment indicators are appropriate expression in the keywords. Urban vibrancy and urban eco-environment are OK.
(2) Page 3. Study area and data source are supposed to be the content of the chapter 2.1. However, the data source has not been presented adequately. In the meantime, the details of the data pre-processing and the assessing unit should be provided.
(3) Many figures and tables are not mentioned in the main text. Please add citation in the main text.
(4) Page 4. In Line 170, there should be reference about Shannon entropy method.
(5) In line 223, the dependent variables are seemed to be eco-environment values.
(6) There are two “figure 2” in page 8.
(7) Figure 4 is not easy to understand. The sankey diagram is usually used to display flows. A simple chart is fine here.
(8) In line 316, the words missiong data and overly seem wrong.
Author Response
Thank you for your comments. The detailed reply is attached for your reference.

Reviewer 2 Report
- The selected research area is very representative in terms of epidemic situations and lifestyle changes. The Introduction is written well, however, I found that the study was only related to the former part of the Introduction section.
- The data for exploring the vitality of cities is also comprehensive with location-based data, social network data, and human mobility data.
Some future thoughts:
- Line 80-81 and some other statements relate to the UHI, I recommend the authors cite studies wordwide. For instance:
Zhao, C., Jensen, J., Weng, Q., Currit, N., & Weaver, R. (2020). Use of Local Climate Zones to investigate surface urban heat islands in Texas, GIScience & Remote Sensing, 57:8, 1083-1101.
Zhao, C., Jensen, J., Weng, Q., Currit, N., & Weaver, R. (2019). Application of airborne remote sensing data on mapping local climate zones: Cases of three metropolitan areas of Texas, US. Computers, Environment and Urban Systems, 74, 175-193. - For the Weibo data, but I feel that Weibo is used by some young people or people with higher levels of knowledge.
- In terms of urban vibrancy indicators, there is no subway data to represent the accessibility.
Author Response

(The authors gave the same response as above.)

Reviewer 3 Report
The manuscript's methodology, structure, and clarity are well-organized and informative to the reader. Wuhan was chosen as the study case in this paper, which makes it an interesting paper. As we know, in the early days of the outbreak, Wuhan was the first city to identify the COVID-19 virus.
I proposed to improve the quality of this research by adding more details about urban plans in relation to this outbreak. Identify the critical mistake of this current situation in relation to urban planning. This is especially important during public lockdown.
This adaptation and new urban plan can provide us with valuable insights. The direct comparison and suggestion may also be used as a recommendation for other countries that use lock down as a method to stop the spread of COVID-19.
Author Response

(The authors gave the same response as above.)

Reviewer 4 Report
Parts of the concept of this paper are promising. New quantitative measures of abstract notions like “vibrancy” could be helpful to urban planners interested in intentionally designing places to have such seemingly ineffable qualities. But the paper overall has major problems in presentation and construction. Also, the second part of the study, which is interested in pollution and surface temperature, sounded interesting, but seems hastily conceived and is not very well linked to the first half. Some more comments follow:
A definition of urban vibrancy would help. How do these authors understand it, and more importantly, why is it important?
Terms like “urban village” and “new urban community” are used throughout but are never defined.
Section 2.2, which introduces the data and method is very difficult to follow, especially on page 4. The equations are meaningless. It tells the reader nothing that DV (density) “is a function of” population density, building density, and so on. What kind of function? Were the component parts just added up? Why?
There are major issues with English usage, grammar, and spelling. This frequently interferes with comprehension. But there are also places where the grammar appears to be correct, but the reasoning underlying the sentence is bewildering. For example, this makes no sense: “The ultimate urban vibrancy value demonstrates the size of the community neighborhood” (p. 4).
The maps seem to be lo-res. Replace.
In the results, the PM2.5 values make no sense if we do not know what baseline healthy numbers would be (p. 9).
Author Response

(The authors gave the same response as above.)

Round 2
Reviewer 3 Report
Overall, I feel the manuscript has been much improved through the author’s revisions.
Author Response
Thank you for the comments. The whole manuscript has been proofread by the native speaker and substantial expressions have been revised accordingly.